# Community Occupational Therapy for people with dementia and family carers (COTiD-UK) versus treatment as usual (Valuing Active Life in Dementia [VALID]) study: A single-blind, randomised controlled trial

Jennifer Wenborn[1,2]*, Aidan G. O'Keeffe[3,4], Gail Mountain[5,6], Esme Moniz-Cook[7], Michael King[1,4], Rumana Z. Omar[3,4], Jacqueline Mundy[8], Jane Burgess[2], Fiona Poland[9], Stephen Morris[10], Elena Pizzo[11], Myrra Vernooij-Dassen[12], David Challis[13], Susan Michie[14], Ian Russell[15], Catherine Sackley[16], Maud Graff[12], Tom Swinson[17], Nadia Crellin[2], Sinéad Hynes[18], Jacki Stansfeld[1,2], Martin Orrell[13]

1 Division of Psychiatry, University College London, London, United Kingdom, 2 Research & Development Department, North East London NHS Foundation Trust, London, United Kingdom, 3 Department of Statistical Science, University College London, London, United Kingdom, 4 Priment Clinical Trials Unit, University College London, London, United Kingdom, 5 School of Health and Related Research (ScHARR), The University of Sheffield, Sheffield, United Kingdom, 6 Centre for Applied Dementia Studies, Faculty of Health Studies, University of Bradford, Bradford, United Kingdom, 7 Faculty of Health Sciences, School of Health & Social Care, University of Hull, Hull, United Kingdom, 8 Essex Stroke Hub Team, North East London NHS Foundation Trust, London, United Kingdom, 9 School of Health Sciences, University of East Anglia, Norwich, United Kingdom, 10 Department of Public Health and Primary Care, University of Cambridge, Cambridge, United Kingdom, 11 Department of Applied Health Research, University College London, London, United Kingdom, 12 Faculty of Medical Sciences, Radboud University Medical Center (Radboudumc), Nijmegen, the Netherlands, 13 Institute of Mental Health, University of Nottingham, Nottingham, United Kingdom, 14 UCL Centre for Behaviour Change, Department of Clinical, Educational and Health Psychology, University College London, London, United Kingdom, 15 Medical School, Swansea University, Swansea, United Kingdom, 16 Department of Public Health Sciences, King's College London, London, United Kingdom, 17 East Herts and Broxbourne Adult Disability Team, Hertfordshire County Council, Stevenage, United Kingdom, 18 School of Health Sciences, National University of Ireland, Galway, Ireland

* j.wenborn@ucl.ac.uk

**Data Availability Statement:** The Trial Outcomes Data, Dataset Guide, and Statistical Analysis Plan

## Abstract

### Background

We aimed to estimate the clinical effectiveness of Community Occupational Therapy for people with dementia and family carers–UK version (Community Occupational Therapy in Dementia–UK version [COTiD-UK]) relative to treatment as usual (TAU). We hypothesised that COTiD-UK would improve the ability of people with dementia to perform activities of daily living (ADL), and family carers' sense of competence, compared with TAU.

### Methods and findings

The study design was a multicentre, 2-arm, parallel-group, assessor-masked, individually randomised controlled trial (RCT) with internal pilot. It was conducted in 15 sites

(SAP) are available from the UCL Research Data Repository (https://rdr.ucl.ac.uk). https://doi.org/10.5522/04/13084151.

**Funding:** This study is funded by the National Institute for Health Research (NIHR) [Programme Grants for Applied Research (Grant Reference Number: RP-PG-0610-10108)], awarded to MO (lead applicant), and DC, MK, SMi, SMo, EM-C, GM, RO, FP, IR, CS, MV-D, JW as co-applicants. URL: www.nihr.ac.uk EP was partially funded by the NIHR Applied Research Collaboration (ARC) North Thames. The funders had no role in study design, data collection and analysis, decision to publish, or preparation of the manuscript.

**Competing interests:** I have read the journal's policy and the authors of this manuscript have the following competing interests: work funded through NIHR research grant to MO.

**Abbreviations:** ACCEPT, Acceptance Checklist for Clinical Effectiveness Pilot Trial; ADL, activities of daily living; BADLS, Bristol Activities of Daily Living Scale; CONSORT, Consolidated Standards of Reporting Trials; COPM, Canadian Occupational Performance Measure; COTiD, Community Occupational Therapy in Dementia; COTiD-UK, Community Occupational Therapy in Dementia–UK version; CRN, Clinical Research Network; CSDD, Cornell Scale for Depression in Dementia; DEMQOL, Dementia Quality of Life Scale; DMEC, Data Monitoring and Ethics Committee; HADS, Hospital Anxiety and Depression Scale; ICC, intraclass correlation coefficient; IDDD, Interview of Deterioration in Daily activities in Dementia; MCID, minimum clinically important difference; MMSE, Mini Mental State Examination; NELFT, North East London NHS Foundation Trust; NIHR, National Institute for Health Research; PROM, patient related outcome measure; PSC, Programme Steering Committee; RCT, randomised controlled trial; SAE, serious adverse event; SCQ, Sense of Competence Questionnaire; TAU, treatment as usual; VALID, Valuing Active Life in Dementia.

across England from September 2014 to January 2018. People with a diagnosis of mild to moderate dementia living in their own home were recruited in pairs with a family carer who provided domestic or personal support for at least 4 hours per week. Pairs were randomised to either receive COTiD-UK, which comprised 10 hours of occupational therapy delivered over 10 weeks in the person with dementia's home or TAU, which comprised the usual local service provision that may or may not include standard occupational therapy. The primary outcome was the Bristol Activities of Daily Living Scale (BADLS) score at 26 weeks. Secondary outcomes for the person with dementia included the following: the BADLS scores at 52 and 78 weeks, cognition, quality of life, and mood; and for the family carer: sense of competence and mood; plus the number of social contacts and leisure activities for both partners. Participants were analysed by treatment allocated. A total of 468 pairs were recruited: people with dementia ranged from 55 to 97 years with a mean age of 78.6 and family carers ranged from 29 to 94 with a mean of 69.1 years. Of the people with dementia, 74.8% were married and 19.2% lived alone. Of the family carers, 72.6% were spouses, and 22.2% were adult children. On randomisation, 249 pairs were assigned to COTiD-UK (62% people with dementia and 23% carers were male) and 219 to TAU (52% people with dementia and 32% carers were male). At the 26 weeks follow-up, data were available for 364 pairs (77.8%). The BADLS score at 26 weeks did not differ significantly between groups (adjusted mean difference estimate 0.35, 95% CI −0.81 to 1.51; $p = 0.55$). Secondary outcomes did not differ between the groups. In total, 91% of the activity-based goals set by the pairs taking part in the COTiD-UK intervention were fully or partially achieved by the final COTiD-UK session. Study limitations include the following: Intervention fidelity was moderate but varied across and within sites, and the reliance on primarily proxy data focused on measuring the level of functional or cognitive impairment which may not truly reflect the actual performance and views of the person living with dementia.

## Conclusions

Providing community occupational therapy as delivered in this study did not improve ADL performance, cognition, quality of life, or mood in people with dementia nor sense of competence or mood in family carers. Future research should consider measuring person-centred outcomes that are more meaningful and closely aligned to participants' priorities, such as goal achievement or the quantity and quality of activity engagement and participation.

## Trial Registration

Current Controlled Trials ISRCTN10748953.

## Author summary

### Why was this study done?

- Maintaining everyday and pleasurable activities can be difficult for a person with dementia, as well as their family carer who often feels increasingly stressed as they need to give more and more support.

- Occupational therapists assist people to improve their health and well-being through carrying on with the activities they need and want to do. It is therefore important to develop effective strategies to deliver occupational therapy to people with dementia and their family carers who support them.

### What did the researchers do and find?

- We tested the effectiveness of the Community Occupational Therapy in Dementia–UK version (COTiD-UK) programme compared to the care that people with mild to moderate dementia and their family carers usually receive, through a clinical trial.

- We recruited 468 pairs comprising a person with dementia and a family carer across England who were randomly allocated to either take part in the COTiD-UK programme or to continue with their usual care.

- We found no statistical evidence that COTiD-UK gave more benefit to people with dementia or their carers than the usual care provided in terms of the people with dementia being able to carry out activities or their mood or quality of life nor their family carers' sense of competence or mood.

- The pairs who took part in the COTiD-UK intervention set an average of 4.09 goals each, of which 91% were fully or partially achieved by the final COTiD-UK session.

### What do these findings mean?

- We did not find evidence to support commissioning the COTiD-UK intervention for people with mild to moderate dementia and their family carers.

- Future studies need to develop more realistic methods of measuring the effectiveness of programmes such as this, particularly to focus on the outcomes that most matter to the individuals taking part and to truly reflect the actual performance, experience, and views of the person living with dementia.

## Introduction

Personalised interventions provided to people with dementia and their family carers either separately or together can improve family carers' well-being, delay admission to care homes, and reduce the risk of institutionalisation by up to one-third [1–3]. In the Netherlands, a Community Occupational Therapy in Dementia (COTiD) intervention for people with mild to moderate dementia and their family caregiver was developed, comprising ten 1-hour sessions

delivered over 5 weeks in the person with dementia's home [4]. A single-site randomised controlled trial (RCT) comparing COTiD with treatment as usual (TAU) demonstrated benefits to activities of daily living (ADL) for people with dementia and improved sense of competence in family carers [5]. The secondary outcomes, quality of life, and mood for people with dementia and their family carers also improved [6], and COTiD was cost-effective [7]. In contrast, a subsequent multisite RCT of COTiD in Germany found no difference in ADL performance between providing COTiD or a single consultation visit by an occupational therapist [8]. The integral process evaluation highlighted the need to not only translate but also to adapt complex interventions to the local context for evaluation and cross-national comparison to be effective [9]. These contrasting findings showed the need for a large multisite RCT of COTiD compared with usual care before it could be adopted into practice in the United Kingdom. We therefore translated the COTiD intervention and adapted it to make it more suitable for the UK using a range of activities. We trained 44 occupational therapists who put COTiD into practice with 130 pairs; ran focus groups and a consensus event; and conducted a national online survey to scope current UK occupational therapy practice and service provision for people with mild to moderate dementia and their family carers living in the community [10–11]. Our adaptations included the following: retaining the 10 hours of intervention but extending the length of delivery to 10 weeks, with individual session length being flexible so as to maximise the achievement of the goals set (for example, to accommodate activities being completed in the community that may take longer than 1 hour) and using a wider range of validated assessment tools in line with usual UK practice. This produced the "Community Occupational Therapy in Dementia–UK version" (COTiD-UK) in readiness for evaluation through an RCT, as part of a funded programme of research "Valuing Active Life in Dementia" (VALID).

Our aim was to estimate the clinical and cost-effectiveness of COTiD-UK relative to TAU. We hypothesised that, in comparison with TAU, COTiD-UK would improve personal care and instrumental ADL performance in people with mild to moderate dementia measured using the Bristol Activities of Daily Living Scale (BADLS) score at 26 weeks as our primary outcome, as well as sense of competence and mood in family carers, and cognition, mood, and quality of life for people with dementia and their family carers as secondary outcomes.

## Methods

This study is reported according to the Consolidated Standards of Reporting Trials (CONSORT) Statement [12]. The CONSORT flowchart is included, and the completed CONSORT checklist is included (S1 CONSORT Checklist).

### Study design

We conducted a multicentre, pragmatic, 2-arm, parallel group, single-blind, individually randomised RCT, with an internal pilot and nested economic evaluation and qualitative and fidelity studies. We have previously published the full study protocol [13]. The economic evaluation and qualitative and fidelity studies are or will be reported separately [14–15].

### Participants

We recruited pairs comprising a person with dementia and their family carer. The former had to live in their own home; have a diagnosis of dementia as defined by the DSM-IV [16]; and score between 0.5 and 2 on the Clinical Dementia Rating Scale indicating mild to moderate dementia [17]. Carers had to be aged 18 or over; and provide practical support with domestic or personal activities to the person with dementia for at least 4 hours per week. Both parties

had to be able to converse in English; be willing to participate in the COTiD-UK intervention together if allocated to receive it; and have the capacity to provide consent.

## Recruitment

**Sites and occupational therapists.** We invited a range of inner city, urban, and rural sites to participate through professional and research networks, including the UK National Institute for Health Research (NIHR) Clinical Research Network (CRN). To take part, organisations had to provide services to people with mild to moderate dementia living in the community and employ occupational therapists who could receive training and then deliver the COTi-D-UK intervention. Potential sites completed a feasibility checklist to confirm that they had these resources.

**People with dementia and their carers.** We approached people with dementia and their carers through NHS clinical services, mainly memory services, and voluntary and charitable bodies providing information and support to people with dementia and their families. We included the study on the Join Dementia Research portal, an online resource that enables people to register their interest in taking part in dementia research and thereby be contacted by relevant study teams (https://www.joindementiaresearch.nihr.ac.uk/). Site-based research staff then contacted interested people, usually by telephone, provided Participant Information Sheets, and followed a defined screening procedure to check eligibility formally. If pairs were eligible and wanted to take part, research staff completed study enrolment by visiting the home of the person with dementia to seek informed consent from both the person with dementia and their family carer.

## Ethical approval

The study was approved by the NHS NRES Committee London–Camberwell St Giles (reference number 14/LO/0736) in June 2014 and local NHS research and development departments at each participating site. Written informed consent was received from all participants, both people with dementia and their family carers.

## Randomisation and masking

We allocated pairs between COTiD-UK and TAU using a remote online randomisation system, stratified by site using random permuted blocks (block sizes of 2, 4, 7, and 14) with study staff masked to block size. An unmasked team member at each site conducted the randomisation, informed pairs of their allocation by letter, and passed their details to the occupational therapist to initiate contact and the intervention.

We masked the following to participants' allocation: all research staff collecting outcome measures; the Chief Investigator, statisticians, health economists, and other collaborators; and the independent Programme Steering Committee (PSC), and Data Monitoring and Ethics Committee (DMEC). It was not possible, nor desirable in a pragmatic trial, to mask the pairs, or the occupational therapists providing the intervention. Precautions to minimise the risk of unmasking, particularly research staff collecting follow-up data, included the following: asking participants not to tell research staff whether or not they had received COTiD-UK; minimising contact between masked staff and occupational therapists; and giving all research staff individual identifiers that provided access only to the necessary sections of the web-based database [18]. We assessed the extent to which research staff had maintained masking by asking them to state after each follow-up visit whether they had been told or inferred the pair's allocation.

## Procedures

### Interventions

**Community Occupational Therapy in Dementia–UK version (COTiD-UK).** COTiD-UK consists of up to 10 hours of community occupational therapy delivered to a person with mild to moderate dementia and their family carer as a pair over 10 weeks. The 10 hours is provided flexibly in response to the pair's availability and to maximise achieving their goals rather than being a set number of sessions. In the first phase, the occupational therapist conducts a one-to-one narrative interview with the 2 participants separately; assesses the home environment; and observes the person with dementia completing a familiar activity of their own choice. The next phase is goal setting, when the occupational therapist summarises the information collected and facilitates a discussion with the pair to enable them to identify, agree, and prioritise individual and joint lifestyle goals. During the intervention phase, the therapist supports the pair to enact their goals and coaches the carer to develop problem-solving skills and coping strategies. The sessions usually take place where the person with dementia lives but depending on the activities chosen, may also happen in the local community, for example, the sports club, cinema, or garden centre. At the final session, the pair and therapist evaluate their success in achieving their goals and plan their future lifestyle accordingly.

Occupational therapists attended 2 consecutive training days and a later follow-up day, delivered by a COTiD-UK trainer—an occupational therapist researcher with experience of developing or delivering COTiD-UK. They then delivered COTiD-UK to a pair recruited solely for training. Therapists audio-recorded these training sessions, as a basis for one-to-one feedback from a COTiD-UK trainer who used a predefined checklist to assess their competency to deliver the intervention in accordance with the study protocol. Participating therapists received monthly supervision throughout their involvement. The COTiD-UK supervisors met monthly by telephone with a COTiD-UK trainer to monitor progress.

Participating therapists audio-recorded all sessions as practicable, to enable researchers to monitor intervention fidelity through a nested longitudinal observational study [15]. We purposively sampled 10% of recorded sessions to include a range of sites and level of therapist experience in delivering COTiD-UK and then transcribed and coded them using fidelity checklists.

We defined adherence to COTiD-UK as completing the goal-setting phase and monitored this using a COTiD-UK Checklist that was developed for the study and completed by the occupational therapists for each pair. The research team and COTiD-UK trainers agreed that this criterion indicated delivery of the initial core elements of the intervention.

**Treatment as usual (TAU).** TAU comprises the usual service provided locally, which could have included standard occupational therapy. As services available to people with dementia and their carers varied between and within sites, each site completed a template detailing usual treatment. To reduce contamination between the 2 groups, we asked occupational therapists trained to deliver COTiD-UK not to provide occupational therapy to trial pairs allocated to TAU nor to share their COTiD-UK training or materials with occupational therapists not so trained. Such limitations on sharing were feasible because COTiD-UK differs in content and duration from current UK occupational therapy practice [11]. At each site, an unmasked researcher monitored contamination by checking whether TAU pairs reported contact with COTiD-UK trained occupational therapists.

### Data collection

We collected the complete dataset through interviews in participants' homes at baseline, 12 and 26 weeks. We collected a reduced dataset from the carer by telephone at 52 and 78 weeks

after randomisation. We planned to follow-up all pairs at 52 weeks and for the first 40% of pairs recruited to the trial at 78 weeks. However, due to delays in recruiting the research sites and then the pairs, alongside the need to complete the trial on time as agreed with the study funder, in practice this was not feasible and was revised.

## Outcomes

The primary outcome measure was total score on the BADLS at 26 weeks [19]. BADLS is a carer-rated scale of ADL performance of 20 personal care and instrumental activities, with higher scores indicating higher dependence. It is valid, reliable, and responsive to change over time [20].

Secondary outcome measures for the person with dementia included cognition (Mini Mental State Examination, MMSE) [21]; quality of life (Dementia Quality of Life Scale, DEMQOL) [22]; assistance needed with ADL (Interview of Deterioration in Daily activities in Dementia, IDDD) [23]; and mood (Cornell Scale for Depression in Dementia, CSDD) [24]. Carer secondary outcomes were sense of competence (Sense of Competence Questionnaire, SCQ) [25] and mood (Hospital Anxiety and Depression Scale, HADS) [26]. We collected the number of social contacts and leisure activities over the previous 12 weeks as a measure of social functioning for both types of participants. For the intervention group only, occupational therapists used the COTiD-UK Checklist to record the number, duration, and content of sessions delivered; the number of goals set; and the number fully or partially achieved.

## Serious adverse events

In accordance with Good Clinical Practice, we recorded deaths and serious adverse events (SAEs), namely those that were life threatening, required or extended hospitalisation, resulted in disability or incapacity, or were otherwise considered significant, as well as vulnerable adult protection alerts.

## Statistical analysis

**Sample size.** We based the target sample size on a standardised mean difference of 0.35 between the COTiD-UK and control groups for the primary outcome of total BADLS score at week 26. The effect size of 0.35 was determined using the clinical expertise of the applicant group and based on the DOMINO group consensus advice regarding the minimum clinically important difference (MCID) using the BADLS [27]. As such, we took a more conservative approach than the findings of Graff and colleagues who demonstrated an effect size of 1.25, albeit in a single-site trial with the intervention delivered by 2 therapists and using a different outcome measure [5].

To detect this, difference with 90% power and a 5% significance level using a 2-sample $t$ test needed 172 pairs in each group, as estimated using STATA version 11 (StataCorp, Texas, United States of America). The total sample size (for both groups) was inflated to allow for 15% attrition and 5% nonadherence at week 26. Additionally, we increased the size of the intervention group to account for within-therapist clustering, assuming an intraclass correlation coefficient (ICC) of 0.015 and an average of 10 pairs per therapist. In total, we planned to recruit 256 pairs to the intervention group and 224 pairs to the TAU group.

**Statistical methods.** We conducted the statistical analysis according to a prespecified plan (S1 Statistical Analysis Plan). We used a linear mixed effects model to compare the COTiD-UK and TAU groups on the total BADLS score at week 26 adjusting for site and baseline total BADLS score (as fixed effects); we included a random effect to account for clustering by occupational therapist in the COTiD-UK arm and estimated separate variance parameters for the

pair-level error terms in COTiD-UK and TAU arms, owing to the more flexible nature of this model compared to that where a common variance parameter would be assumed for both arms.

For continuous secondary outcomes, we fitted similar models, adjusting for baseline values of the outcome, site, and a random effect to account for clustering by occupational therapist. We used negative binomial regression models to analyse numbers of social contacts and leisure activities. We included only participants for whom total BADLS scores were available at both baseline and week 26 in these regression analyses.

We conducted 3 supportive analyses for the primary outcome. First, we analysed the within-pair mean BADLS score at week 26, i.e., the within-pair sample mean score for complete answers on the BADLS questionnaire, to account for missing items. Secondly, we analysed total BADLS score at week 26, adjusted for baseline predictors of missing data. We identified predictors of missing data using logistic regression models; variables that were significant ($p$-value $< 0.05$) were included as predictors of missing data. Thirdly, we estimated the complier average causal effect of COTiD-UK on total BADLS score at week 26, using a 2-stage least squares approach to account for pairs who did not adhere with COTiD-UK in that they did not complete the goal-setting phase; however, it was not possible to account for clustering by occupational therapist in this model.

We estimated the longer-term intervention effect of COTiD-UK using a mixed effects regression model with time as an explanatory variable and using repeated measures of BADLS at weeks 26, 52, and 78, and similar models for the HADS anxiety and depression scores also recorded at weeks 52 and 78. All these analyses were performed on participants as randomised using STATA version 15.

## Trial management

The NIHR appointed an independent DMEC to oversee the trial conduct and the safety of participants. This committee received Open and Closed reports with the latter remaining confidential to the Trial Manager and independent DMEC members and was not available to the Trial Statistician who attended a portion of each meeting. Baseline, reason for withdrawal, and SAE data were provided by group allocated in the Closed report. All data in both reports were anonymised, and no outcome data were included. The DMEC reported to the independent PSC, also appointed by the NIHR. North East London NHS Foundation Trust (NELFT) was the study sponsor and carried out the necessary oversight and monitoring to ensure the study was conducted in line with all mandatory governance requirements.

We registered the trial with Current Controlled Trials as ISRCTN10748953.

The internal pilot ran from September 2014 to April 2015 in 3 sites. We reviewed the procedures and dataset against a predefined Acceptance Checklist for Clinical Effectiveness Pilot Trial (ACCEPT) and presented this to the PSC in April 2015 [28]. The clinical researcher responsible for training and supervising the occupational therapists delivering the intervention was unmasked to allocation and so reviewed the SAEs reported to ensure participant safety and the COTiD-UK Checklist data to check accuracy and to monitor intervention fidelity, while all other data were reviewed by masked team members. No individual participant data were presented to the PSC. Participant recruitment had been slower than expected but was picking up, and we had achieved 88% retention at 12 weeks at the first site. We therefore decided that we needed to recruit more sites and occupational therapists than originally planned in order to enable us to recruit the required sample to time and target. During the pilot, we completed the Canadian Occupational Performance Measure (COPM) [29] but decided to omit this from the main trial, partly because we felt we needed to reduce the time

needed for data collection to lessen the burden on participants, partly due to the feedback from research staff about the difficulties experienced in administering this assessment in a consistent and reliable way over time and between assessors, and partly due to our concern that its completion involved participants setting activity goals that constituted an element of the COTiD-UK intervention which made it unsuitable for use with the TAU group. Other than this change, we confirmed that the design and methods were feasible, and the PSC supported the internal pilot data being included within the full trial dataset. The results we present here are therefore derived from the dataset collected during the internal pilot and full RCT.

## Results

We recruited 15 sites between 22 September 2014 and 17 May 2017. One site did not proceed to recruiting pairs due to a major service reorganisation resulting in the occupational therapists we had trained not being available to take part. We recruited pairs of participants within the other 14 sites between 30 September 2014 and 3 July 2017; the final follow-up assessment was on 23 January 2018 when the study shut to recruitment. We trained 44 occupational therapists to deliver COTiD-UK, of whom 32 proceeded to the RCT and were allocated at least 1 pair each, although 1 was subsequently unavailable to provide the intervention as planned due to ill health.

Fig 1 shows the flow of pairs through the trial.

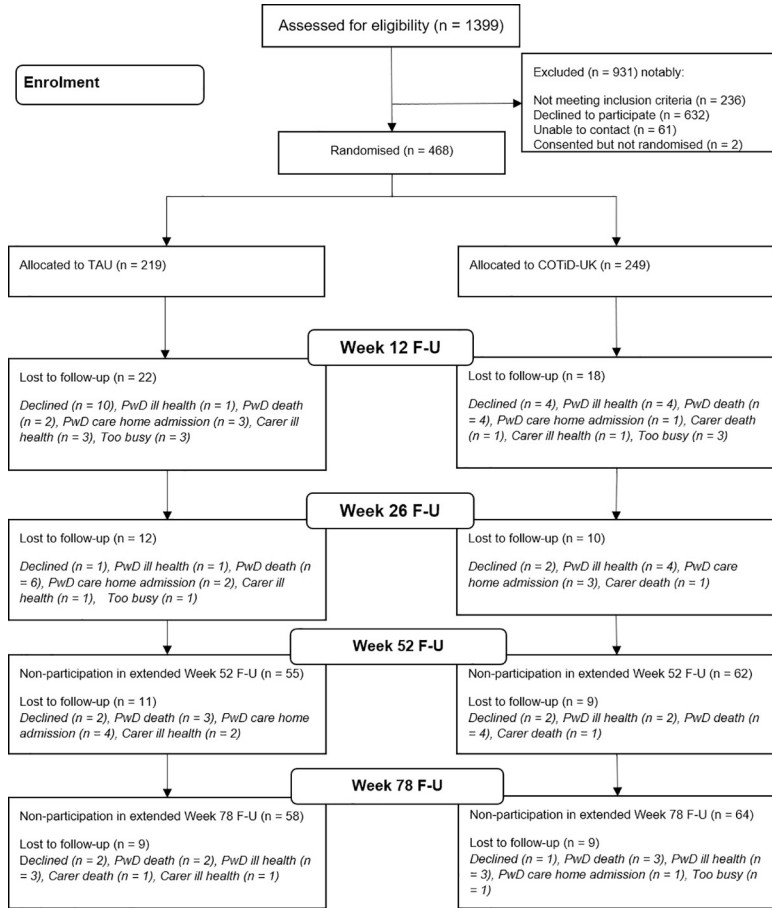

**Fig 1. Participant flowchart.** COTiD-UK, Community Occupational Therapy in Dementia–UK version; PwD, person with dementia; TAU, treatment as usual.

We allocated 468 pairs at random: 249 to COTiD-UK and 219 to TAU. Baseline characteristics were very similar between the 2 groups (Table 1). People with dementia ranged in age from 55 to 97 with a mean of 78.6 years, and family carers ranged in age from 29 to 94 with a mean of 69.1 years. Of the people with dementia, 3 quarters were married (350/468, 74.8%), and approximately one-fifth (90/468, 19.2%) lived alone. Approximately 3 quarters of the family carers were spouses, and the majority of the remainder were adult children.

We collected outcome data for 400 pairs (85.5% of the original sample) at 12 weeks (COTiD-UK $n$ = 216; TAU $n$ = 184) and then for 368 (78.6% of sample) at 26 weeks (COTiD-UK $n$ = 207; TAU $n$ = 161). At 26 weeks, 406 pairs (86.7% of sample: COTiD-UK $n$ = 221; TAU $n$ = 185) remained in the trial. Of these, we analysed total BADLS score, the primary outcome, for 364 pairs (77.8% of sample) (206 in the COTiD-UK group and 158 in the control group) after excluding 4 pairs for whom a total BADLS score at baseline was missing.

At 52 weeks, we followed up 292 pairs (62.4% of sample) and of these, collected and analysed total BADLS outcome data for 239 pairs (51.1% of sample: COTiD-UK $n$ = 137; TAU $n$ = 102). At 78 weeks, we followed up 150 pairs (32.1% of sample) and of these, collected and analysed total BADLS outcome data for 113 pairs (24.1% of the original sample: COTiD-UK $n$ = 67; TAU $n$ = 46). Total BADLS scores at 26 weeks did not differ at the 5% level between the randomised groups: The adjusted mean difference estimate was 0.35 (95% CI −0.81 to +1.51; $p$ = 0.55). Though staff collecting outcome data could deduce the allocated treatment for 45 (13%) of the 338 assessments, there is no evidence that this influenced scores. The adjusted (for baseline total BADLS total and randomised group) ICC for the primary outcome at 26 weeks was 0.043.

Results from the supportive analyses were consistent with those for the primary outcome analysis. When considering within-pair mean BADLS score, the mean difference estimate was 0.02 (95% CI −0.04 to +0.07). Predictors of missingness were identified as ethnicity of the person with dementia (white, other, or unspecified) and marital status of the family carer (married, widowed, or other). When adjusting for these predictors, the mean difference estimate in total BADLS scores was 0.34 (95% CI −0.82 to +1.49). The estimated complier average causal effect of COTiD-UK at 26 weeks was 0.42 (95% CI −0.77 to +1.60).

Secondary outcomes were also similar between the groups at 26 weeks (Table 2). For continuous outcomes, all estimated effects were close to 0. For leisure and social contacts, intensity rate ratio estimates were close to 1.

Fig 2 and Table 3 show total BADLS scores at 52 and 72 weeks by randomised groups. Total BADLS scores did not differ significantly between groups across time points in an adjusted longitudinal analysis: The adjusted mean difference estimate was 0.49 (95% CI −0.47 to +1.45). Similarly, the HADS anxiety and depression scores did not differ between groups at weeks 52 and 78 (Figs 3 and 4).

Between baseline and 26 weeks researchers and therapists recorded 131 SAEs, but did not attribute any in the intervention group to COTiD-UK participation or any in the TAU group to its absence.

Of the 249 pairs allocated to COTiD-UK, 22 (9%) did not reach the goal-setting phase. Excluding 3 deaths and 8 withdrawals, the nonadherence rate was 4.6%, very close to our prior estimate of 5%. We assessed intervention fidelity as being moderate, albeit with variation between sites and therapists [15]. We were successful in avoiding occupational therapists trained in COTiD-UK from providing any intervention to TAU participants. Of the 227 adherent pairs, 2 did not set any goals, and the remainder set a mean of 4.09 goals (range 1 to 13). Of the 920 goals set, 729 (79.2%) were fully achieved, and a further 107 (11.6%) were partially achieved, making a total of 90.8% goals fully or partially achieved. In the main, pairs set goals related to resuming or continuing with meaningful activities, developing strategies to

**Table 1. Participant characteristics at baseline by allocated group.**

| Numbers (%) unless otherwise specified | Person with dementia | | Family carer | |
|---|---|---|---|---|
| | COTiD-UK (*n* = 249) | TAU (*n* = 219) | COTiD-UK (*n* = 249) | TAU (*n* = 219) |
| **Gender** | | | | |
| Male | 154 (62%) | 113 (52%) | 58 (23%) | 71 (32%) |
| Female | 95 (38%) | 106 (48%) | 191 (77%) | 148 (68%) |
| **Age** (years) Mean (SD) Median | 78.4 (7.0) 79.4 | 78.8 (7.5) 78.7 | 69.6 (11.6) 71.9 | 68.6 (12.2) 71.6 |
| **Ethnicity** | | | | |
| White British | 227 (91%) | 200 (91%) | 228 (92%) | 202 (92%) |
| White other | 8 (3%) | 10 (5%) | 4 (2%) | 7 (3%) |
| Afro Caribbean | 6 (2%) | 6 (3%) | 9 (4%) | 6 (3%) |
| South Asian | 5 (2%) | 1 (0.5%) | 5 (2%) | 1 (0.5%) |
| Other or no response | 3 (1%) | 2 (1%) | 3 (1%) | 3 (1%) |
| **Marital status** | | | | |
| Married | 191 (77%) | 159 (73%) | 208 (83%) | 185 (84%) |
| Widow(er) | 46 (18%) | 46 (21%) | 8 (3%) | 3 (1%) |
| Other | 12 (5%) | 14 (6%) | 33 (13%) | 31 (14%) |
| **Carer's relationship to person with dementia** | | | | |
| Spouse or partner | - | - | 189 (76%) | 151 (69%) |
| Son or daughter | - | - | 47 (19%) | 57 (26%) |
| Other | - | - | 13 (5%) | 11 (5%) |
| **Living situation of person with dementia** | | | | |
| Living alone | 44 (18%) | 46 (21%) | - | - |
| Living with spouse or partner | 190 (76%) | 162 (74%) | - | - |
| Living with others | 15 (6%) | 11 (5%) | - | - |
| **Dementia diagnosis** | | | | |
| Alzheimer's disease | 132 (53%) | 115 (52%) | - | - |
| Vascular dementia | 47 (19%) | 45 (21%) | - | - |
| Mixed | 30 (12%) | 30 (14%) | - | - |
| Other or not known | 40 (16%) | 29 (13%) | - | - |
| **Clinical Dementia Rating** | | | | |
| 0.5 | 82 (33%) | 74 (34%) | - | - |
| 1.0 | 135 (54%) | 115 (53%) | - | - |
| 2.0 | 32 (13%) | 28 (13%) | - | - |
| **BADLS** | *n* = 248 | *n* = 215 | - | - |
| | 13.6 (9.0) | 14.7 (9.7) | | |
| **MMSE** | *n* = 249 | *n* = 219 | - | - |
| | 20.7 (5.3) | 21.4 (4.8) | | |
| **IDDD** | *n* = 249 | *n* = 218 | - | - |
| | 45.9 (10.5) | 46.9 (11.6) | | |
| **DEMQOL** | *n* = 234 | *n* = 212 | - | - |
| | 92.1 (12.7) | 90.8 (12.4) | | |
| **DEMQOL proxy** | *n* = 238 | *n* = 204 | - | - |
| | 93.4 (12.5) | 93.1 (13.9) | | |
| **CSDD** | *n* = 246 | *n* = 216 | - | - |
| | 4.3 (4.2) | 4.5 (3.8) | | |
| **SCQ** | - | - | *n* = 240 | *n* = 213 |
| | | | 94.6 (17.6) | 93.8 (15.8) |

(*Continued*)

**Table 1.** (Continued)

| Numbers (%) unless otherwise specified | Person with dementia | | Family carer | |
|---|---|---|---|---|
| | COTiD-UK (*n* = 249) | TAU (*n* = 219) | COTiD-UK (*n* = 249) | TAU (*n* = 219) |
| **HADS—anxiety** | - | - | *n* = 248 | *n* = 217 |
| | | | 6.6 (4.5) | 6.8 (4.4) |
| **HADS—depression** | - | - | *n* = 249 | *n* = 217 |
| | | | 4.2 (3.5) | 4.3 (3.3) |
| **Social contacts** | *n* = 245 | *n* = 216 | *n* = 246 | *n* = 217 |
| | 38.1 (29.7) | 40.9 (38.8) | 41.7 (50.7) | 41.9 (34.9) |
| **Leisure activities** | *n* = 246 | *n* = 218 | *n* = 247 | *n* = 218 |
| | 31.7 (38.4) | 33.8 (43.1) | 39.2 (45.8) | 37.9 (43.0) |

BADLS, Bristol Activities of Daily Living Scale; COTiD-UK, Community Occupational Therapy in Dementia–UK version; CSDD, Cornell Scale for Depression in Dementia; DEMQOL, Dementia Quality of Life Scale; HADS, Hospital Anxiety and Depression Scale; IDDD, Interview of Deterioration in Daily activities in Dementia; MMSE, Mini Mental State Examination; SCQ, Sense of Competence Questionnaire; TAU, treatment as usual.

enable living with dementia, and connecting with their local community and potential sources of support. The goals that were partially or not achieved were frequently dependent on other services or community resources for completion so if these were not available or able to respond within the timeframe of the intervention, for example, a community group that met infrequently, some goals were still pending at the time of the evaluation.

**Table 2. Outcomes at 26 weeks for people with dementia and family carers by allocated group.**

| Measure (*n* = COTiD-UK / TAU groups) | Means (standard deviations) | | Adjusted* mean difference (95% CI) |
|---|---|---|---|
| | COTiD-UK | TAU | |
| **People with dementia** | | | |
| **BADLS** (*n* = 207 / 161) | 15.4 (9.6) | 16.5 (10.7) | 0.35 (−0.81, 1.51); *p* = 0.55 |
| **MMSE** (192 / 145) | 20.4 (6.1) | 20.9 (5.4) | 0.35 (−0.40, 1.11) |
| **IDDD** (194 / 150) | 47.2 (12.8) | 47.7 (12.7) | 0.03 (−1.89, 1.96) |
| **DEMQOL** (*n* = 183 / 140) | 93.6 (11.7) | 92.6 (11.7) | 0.51 (−1.36, 2.38) |
| **DEMQOL proxy** (*n* = 182 / 141) | 93.6 (14.5) | 92.4 (13.6) | 0.70 (−1.32, 2.72) |
| **CSDD** (*n* = 190 / 144) | 3.4 (4.0) | 3.6 (4.2) | −0.002 (−0.77, 0.77) |
| **Family carer** | | | |
| **SCQ** (*n* = 180 / 140) | 95.6 (18.3) | 95.0 (15.7) | −0.09 (−2.52, 2.35) |
| **HADS—anxiety** (*n* = 192 / 151) | 6.6 (4.4) | 6.3 (4.1) | 0.39 (−0.16, 0.93) |
| **HADS—depression** (*n* = 192 / 150) | 4.6 (3.6) | 4.5 (3.1) | 0.18 (−0.36, 0.73) |
| | | | Adjusted** intensity rate ratio (95% CI) |
| **PwD social contacts** (*n* = 196 / 152) | 39.9 (38.5) | 42.3 (42.7) | 0.99 (0.83, 1.18) |
| **FC social contacts** (*n* = 195 / 152) | 39.5 (33.7) | 39.0 (38.2) | 1.17 (0.98, 1.41) |
| **PwD leisure activities** (*n* = 196 / 152) | 37.8 (40.6) | 36.2 (41.7) | 1.01 (0.79, 1.29) |
| **FC leisure activities** (*n* = 195 / 151) | 36.8 (42.5) | 36.7 (37.1) | 1.03 (0.82, 1.28) |

*Adjusted for baseline value, site, and occupational therapist clustering in the COTiD-UK arm.

**Adjusted for baseline value and site.

BADLS, Bristol Activities of Daily Living Scale; COTiD-UK, Community Occupational Therapy in Dementia–UK version; CSDD, Cornell Scale for Depression in Dementia; DEMQOL, Dementia Quality of Life Scale; FC, family carer; HADS, Hospital Anxiety and Depression Scale; IDDD, Interview of Deterioration in Daily activities in Dementia; MMSE, Mini Mental State Examination; PwD, person with dementia; SCQ, Sense of Competence Questionnaire; TAU, treatment as usual.

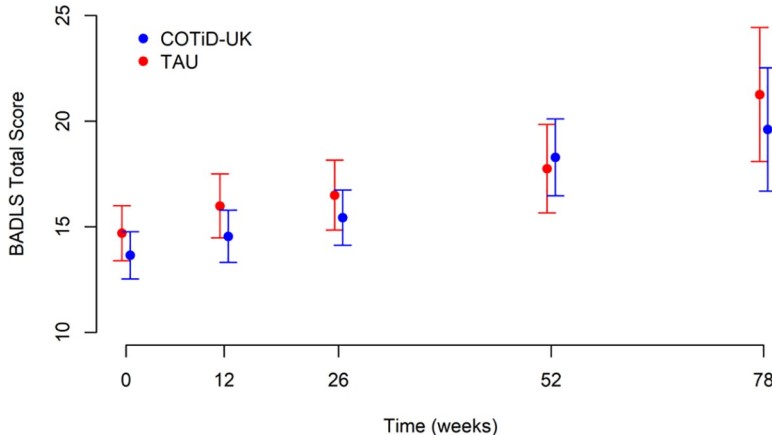

**Fig 2. Total BADLS score: means and 95% CIs over time by allocated group.** BADLS, Bristol Activities of Daily Living Scale; COTiD-UK, Community Occupational Therapy in Dementia–UK version; TAU, treatment as usual.

## Discussion

To our knowledge, this was the biggest RCT of occupational therapy for people with mild to moderate dementia and their family carers to date in the UK, conducted in 15 sites across England. We recruited 468 pairs who were randomly allocated to either the COTiD-UK (249) or TAU (219) group. We observed no statistically significant difference between the COTi-D-UK and TAU groups for the primary outcome, total BADLS score at 26 weeks, or the secondary outcome measures, at any of the 4 follow-up points. Although there were no significant differences in the validated outcome measures, there was a very high rate of goal attainment in the intervention group. Admittedly, we were not able to compare these rates with the TAU group who had not set goals, since goal setting was a key component of the COTiD-UK intervention and could therefore not realistically be completed with the TAU group.

Our results therefore accord with those of the trial of COTiD conducted in Germany [8] but not the Netherlands [5–6]. Usual care differed greatly between the Netherlands and the UK, which may have had an impact on the results. When the Dutch study was conducted, community occupational therapy for people with dementia in the Netherlands was in its infancy. Since the VALID research programme began in 2012, UK memory services have

**Table 3. Outcomes at 52 and 78 weeks for people with dementia and family carers by allocated group.**

| Means (SDs) | 52 weeks | | 78 weeks | |
|---|---|---|---|---|
| Measure | COTiD-UK | TAU | COTiD-UK | TAU |
| **People with dementia** | | | | |
| **BADLS** | 18.3 (10.9) (*n* = 137) | 17.7 (10.8) (*n* = 102) | 19.6 (12.2) (*n* = 67) | 21.3 (11.0) (*n* = 46) |
| **Family carer** | | | | |
| **HADS—anxiety** | 6.4 (4.2) (*n* = 137) | 6.5 (4.3) (*n* = 102) | 5.8 (4.2) (*n* = 63) | 6.6 (4.3) (*n* = 45) |
| **HADS—depression** | 4.6 (3.6) (*n* = 138) | 5.1 (3.9) (*n* = 102) | 4.9 (4.1) (*n* = 64) | 5.7 (3.5) (*n* = 45) |

BADLS, Bristol Activities of Daily Living Scale; COTiD-UK, Community Occupational Therapy in Dementia–UK version; HADS, Hospital Anxiety and Depression Scale; TAU, treatment as usual.

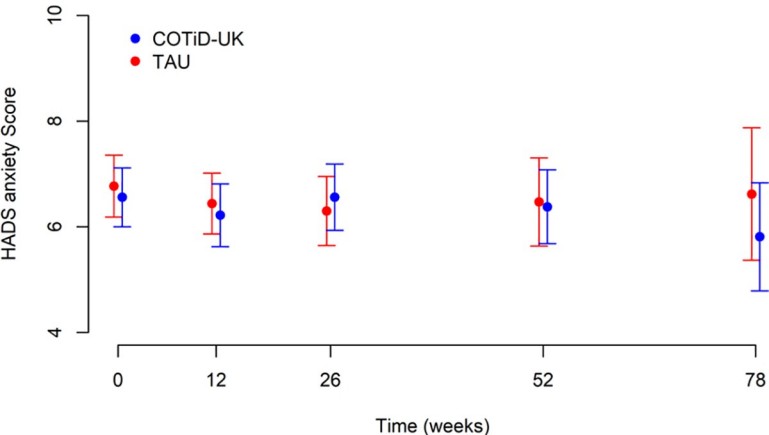

**Fig 3. HADS anxiety score: means and 95% CIs over time by allocated group.** COTiD-UK, Community Occupational Therapy in Dementia–UK version; HADS, Hospital Anxiety and Depression Scale; TAU, treatment as usual.

become well established, and the policy emphasis for earlier diagnosis and access to services means more people may receive occupational therapy input during the mild to moderate stage of dementia, although this is usually less than the COTiD-UK intervention and focused on just the person with dementia's needs rather than being dyadic in nature [11]. Some potential participants declined to take part in the study as the family carers stated they were "managing fine" and therefore did not see the need for intervention [30]. Having said that, 227 of the 249 pairs allocated to take part in the intervention did set goals, of which 91% were fully or partially achieved.

This study had several strengths. Firstly, this was an RCT that achieved 97.5% of the target recruitment sample, as well as low attrition and nonadherence rates, with no evidence of contamination between the 2 groups. Intervention fidelity was moderate suggesting that the participants received the sessions as planned, although there was variation between sites and between occupational therapists. This was lower than that reported by Graff and colleagues in a single-site trial that involved only 2 therapists who were deeply committed and closely

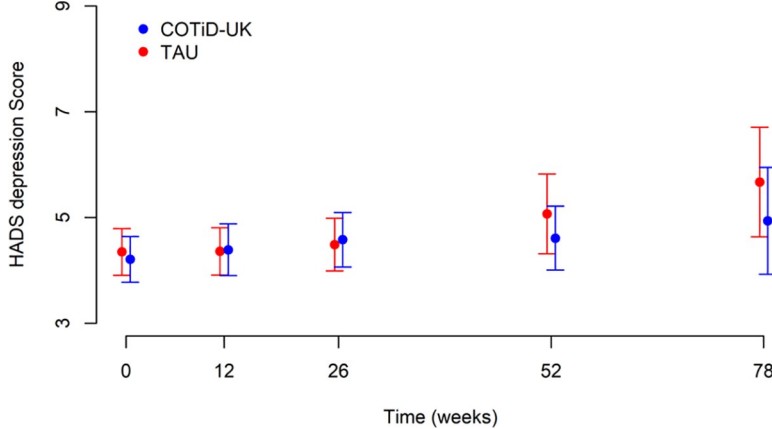

**Fig 4. HADS depression score: means and 95% CIs over time by allocated group.** COTiD-UK, Community Occupational Therapy in Dementia–UK version; HADS, Hospital Anxiety and Depression Scale; TAU, treatment as usual.

monitored [5]. In contrast, our trial was much larger involving 32 occupational therapists across England, whose level of prior dementia and specific COTiD-UK experience varied. In addition, this pragmatic trial of COTiD-UK was conducted across 15 sites and against a range of organisational issues that inevitably had an impact on the delivery of the intervention by the occupational therapists who relied on their managers and colleagues' support to implement this new intervention, often alongside their usual practice. As reported earlier, 1 site was unable to progress to actually recruit pairs due to service reorganisation that resulted in the occupational therapists we had trained no longer being available.

There were also several limitations. Overall, the sample was representative of the target population, i.e., people with mild to moderate dementia and their family carers living in the community, excepting for ethnicity. The eligibility criteria of needing to speak English obviously affected recruitment of participants from non-English-speaking populations and is not unusual [31]. However, this was essential to enable data collection to be completed and the intervention to be provided on the basis of all providers and participants sharing similar levels of first language skills.

There was often an issue of recall for people with dementia. Hence, the majority of outcome data relating to the person with dementia were collected by proxy and may not reflect the actual performance, experience, and views of the person with dementia. An observational ADL measure, such as the Assessment of Motor and Process Skills [32] which Graff and colleagues [5] used might have provided more accurate data about the person with dementia's performance. However, this has to be administered by an occupational therapist who has undergone additional training and achieved calibration. As the percentage of AMPS trained occupational therapists is far lower in the UK than in the Netherlands, this multisite study was not able to adopt this measure due to the resource implications as noted previously [33]. Many participants, both people with dementia and family carers, also found it difficult to recall what leisure and social activities they had done over the preceding 12 weeks and, although it had originally been planned to ask participants to complete a diary tool, evidence indicates that these are rarely completed accurately, and so this plan was dropped. Hence, the accuracy of these data is questionable [34]. Despite the Graff study [5] showing improvement, at least in the short term, there are inherent difficulties in expecting improvements in ADL in someone with a progressive condition such as dementia. Measuring the impact on cognition, ADL, and quality of life was established as a feature of pharmacological trials that has subsequently been adopted when evaluating psychosocial interventions. Perhaps maintenance rather than improved performance would be a more realistic measure of effectiveness. We also noted that in the main, pairs set goals related to a wider range of activities than those covered within a traditional ADL assessment such as the BADLS, for example, leisure, creative, and social activities. The outcome tools primarily measured level of impairment, but there is now increasing emphasis within dementia research on the importance of measuring potentially positive outcomes rather than just simply the negative consequences of dementia, such as cognitive impairment and functional decline. A number of positive psychology measures have now been developed, which may help us to better understand how people's capabilities and self-management skills might be enhanced through training and goal setting [35]. Furthermore, a developing research agenda is focused on developing patient-related outcome measures (PROMs) in formats that make them suitable for self-report. This would enable data to be collected directly from people with dementia themselves to reflect their own views in preference to the traditional reliance on proxy measurement via their carers who obviously provide their own point of view.

There is a need for innovation and improvement in dementia care in all health systems, and this study shows that it is feasible to conduct rigorous clinical trials of complex interventions

by practitioners across many sites. Our occupational therapists recognised the relevance of the intervention to their practice and showed commitment and persistence in delivering the trial.

We found no statistical evidence of clinical benefits for COTiD-UK as delivered in this study. The intervention delivery achieved moderate fidelity, suggesting that people with dementia and family carers received it as planned in terms of its aim and purpose. Trials of psychosocial dementia interventions are still in their infancy so much can be learnt from this study. With both the COTiD-UK and the German COTiD trials not finding significant benefits, it is not clear how far COTiD should be further evaluated using this kind of design, hence alternative designs and outcome measures need to be considered.

## Supporting information

**S1 CONSORT Checklist.**
(DOC)

**S1 Statistical Analysis Plan.**
(PDF)

## Acknowledgments

North East London NHS Foundation Trust (NELFT) sponsored the study.

The authors thank all the people with dementia and their family carers who took part, the occupational therapists who delivered the intervention, and the research staff and clinical teams who recruited participants, collected data, and conducted the study across sites.

The authors also acknowledge the support of the NIHR Comprehensive Clinical Research Network in England, UCLH NIHR Biomedical Research Centre (BRC), UCL Priment Clinical Trials Unit, and Sealed Envelope Ltd.

The views expressed are those of the authors and not necessarily those of the NIHR or the Department of Health and Social Care.

## Author Contributions

**Conceptualization:** Jennifer Wenborn, Gail Mountain, Esme Moniz-Cook, Michael King, Rumana Z. Omar, Fiona Poland, Stephen Morris, Myrra Vernooij-Dassen, David Challis, Susan Michie, Ian Russell, Catherine Sackley, Maud Graff, Martin Orrell.

**Data curation:** Jennifer Wenborn, Aidan G. O'Keeffe, Jacqueline Mundy, Jane Burgess, Elena Pizzo, Tom Swinson, Nadia Crellin, Sinéad Hynes, Jacki Stansfeld.

**Formal analysis:** Aidan G. O'Keeffe, Rumana Z. Omar, Elena Pizzo, Tom Swinson.

**Funding acquisition:** Jennifer Wenborn, Gail Mountain, Esme Moniz-Cook, Michael King, Rumana Z. Omar, Fiona Poland, Stephen Morris, Myrra Vernooij-Dassen, David Challis, Susan Michie, Ian Russell, Catherine Sackley, Martin Orrell.

**Investigation:** Jacqueline Mundy, Jane Burgess, Elena Pizzo, Tom Swinson, Nadia Crellin, Sinéad Hynes, Jacki Stansfeld.

**Methodology:** Jennifer Wenborn, Aidan G. O'Keeffe, Gail Mountain, Esme Moniz-Cook, Michael King, Rumana Z. Omar, Fiona Poland, Stephen Morris, Myrra Vernooij-Dassen, David Challis, Susan Michie, Ian Russell, Catherine Sackley, Maud Graff, Martin Orrell.

**Project administration:** Jennifer Wenborn, Jacqueline Mundy, Jane Burgess, Elena Pizzo, Nadia Crellin, Sinéad Hynes, Martin Orrell.

**Resources:** Maud Graff.

**Supervision:** Jennifer Wenborn, Gail Mountain, Esme Moniz-Cook, Michael King, Rumana Z. Omar, Jacqueline Mundy, Jane Burgess, Nadia Crellin, Sinéad Hynes, Martin Orrell.

**Validation:** Rumana Z. Omar.

**Visualization:** Jennifer Wenborn, Aidan G. O'Keeffe.

**Writing – original draft:** Jennifer Wenborn, Aidan G. O'Keeffe, Gail Mountain, Michael King, Rumana Z. Omar, Fiona Poland, Martin Orrell.

**Writing – review & editing:** Jennifer Wenborn, Aidan G. O'Keeffe, Gail Mountain, Esme Moniz-Cook, Michael King, Rumana Z. Omar, Jacqueline Mundy, Jane Burgess, Fiona Poland, Stephen Morris, Elena Pizzo, Myrra Vernooij-Dassen, David Challis, Susan Michie, Ian Russell, Catherine Sackley, Maud Graff, Tom Swinson, Nadia Crellin, Sinéad Hynes, Jacki Stansfeld, Martin Orrell.

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
