## [Editor Report · Decision Letter 0]

17 Feb 2020

Dear Dr Wenborn, 

Thank you for submitting your manuscript entitled "Community Occupational Therapy for people with dementia and family carers (COTiD-UK) versus treatment as usual (Valuing Active Life in Dementia [VALID]) study: a single-blind, randomised controlled trial" for consideration by PLOS Medicine.

Your manuscript has now been evaluated by the PLOS Medicine editorial staff and I am writing to let you know that we would like to send your submission out for external peer review.

Please re-submit your manuscript within two working days, i.e. by 19 Feb 2020, 11:59PM.

Kind regards,

Louise Gaynor-Brook, MBBS PhD

Associate Editor, PLOS Medicine

---

## [Decision Letter · Decision Letter 1]

29 May 2020

Dear Dr. Wenborn,

Thank you very much for submitting your manuscript "Community Occupational Therapy for people with dementia and family carers (COTiD-UK) versus treatment as usual (Valuing Active Life in Dementia [VALID]) study: a single-blind, randomised controlled trial" (PMEDICINE-D-20-00486R1) for consideration at PLOS Medicine. 

[LINK]

In light of these reviews, I am afraid that we will not be able to accept the manuscript for publication in the journal in its current form, but we would like to consider a revised version that addresses the reviewers' and editors' comments. Obviously we cannot make any decision about publication until we have seen the revised manuscript and your response, and we plan to seek re-review by one or more of the reviewers. 

We expect to receive your revised manuscript by Jun 19 2020 11:59PM. Please email us (plosmedicine@plos.org) if you have any questions or concerns.

We look forward to receiving your revised manuscript. 

Sincerely,

Emma Veitch, PhD

PLOS Medicine

On behalf of Clare Stone, PhD, Acting Chief Editor,

PLOS Medicine

plosmedicine.org

*At this stage, we'd appreciate it if you could structure your abstract using the PLOS Medicine headings (Background, Methods and Findings, Conclusions). The Methods and Findings sections are one combined sub-section, “Methods and findings”. Please also make sure each subsection is written in complete sentence text (ie not sentence fragments).

*In the last sentence of the Abstract Methods and Findings section, please summarise some of the key limitation(s) of the study's methodology.

*At this stage, we ask that you include a short, non-technical Author Summary of your research to make findings accessible to a wide audience that includes both scientists and non-scientists. The Author Summary should immediately follow the Abstract in your revised manuscript. This text is subject to editorial change and should be distinct from the scientific abstract. Please see our author guidelines for more information: https://journals.plos.org/plosmedicine/s/revising-your-manuscript#loc-author-summary

*In some places in the main text, the writeup involves sentence fragments rather than complete sentences (eg, Methods/Study Design section). We'd suggest rewriting these sections so they are in complete sentences.

*The completed CONSORT flowchart and checklist have been included with the paper (as figure/supporting information), which is great, we would just suggest also adding a short statement to the Methods section of the paper that the CONSORT guideline was used to guide reporting of the paper (and then to call out/reference the CONSORT checklist supporting information file). 

Comments from the reviewers:

Reviewer #1: This is a very well written and well conducted study with important findings. 

I was very interested to read the results - Fidelity and the outcome measures used appear to be related to the lack of benefit detected. 

I suggest only two minor revisions

1) Please describe early in the paper which aspect of ADL the Bristol Measure uses (eg IADL, BADL, both, performance of, participation in etc....) as I and other readers will be less familiar with the Bristol measure

2) Can you report on how many consultations the intervention group received (will help with understanding fidelity)

Reviewer #2: Thanks for the opportunity to review your manuscript. My role is as a statistical reviewer and so my comments focus on the study design, data, and analysis of the study (and the presentation of these). I have put general comments, and followed these with specific queries with a page/line reference. 

The methods section mentions a prespecified plan - is this available? It would be very helpful when reviewing the statistical aspects of the manuscript, I think many of my queries below could be resolved with additional information from a SAP. 

The study used a data monitoring and ethics committee to oversee trial conduct and safety (p13, L296), and has also been listed as being masked to the participants allocation. For the DMEC, does the masking refer to access to individual patient data or did the committee only have access to blinded monitoring reports throughout the study?

P6, L140. It wasn't clear to me when I read this paragraph that BADSL was the main outcome of the study and that QoL and the other measures were secondary. 

P7, L152. Was the rationale for excluding patients with more severe dementia because of issues around consent?

P8, L182. What size block was used? Were study staff blinded to the block size? 

P10, L243. To clarify, was the follow-up at 52 weeks ceased once 77% of participants contacted and interviewed? How were the 40% at 78 weeks selected for follow-up? 

P12, L266. What criteria was used to decide if this is a clinically significant change? Was there consumer input into this criteria?

P12, L270. Was the main analysis originalyl planned to be ITT given an adjustment for non-adherence is included? Was the allocation ratio adjusted from 1:1 to account for the 5% non-adherence in the intervention group?

P12, L275. Was the site effect included as a random intercept or a fixed effect?

P13, L302. Was the data from the internal pilot blinded or unblinded? 

P15, L347. The use of these additional variables here wasn't clear (and might be cleared up if the SAP is available) to me. Was the idea to have a main analysis robust to MAR using the mixed-effect model repeated-measures (MRMM), and then extend this to include other factors associated with missingness?

P25, Fig 2. I'd consider including the modelled estimates of BADLS from the linear mixed models (marginal means) in addition to the summary measures, as these will include the adjustment for data that is missing at random used to test for differences between arms.

P31, Tab 3. Which part of the table does the footnote apply to

Reviewer #3: This is an interesting and important paper. The methodology of the research is well described, strong and accurate. Although the results of the research are disappointing, publishing papers on interventions that did not yield the expected results is just as important as publishing about successful interventions. Moreover, possible explanations for the lack of effects are also part of the paper. The paper is well written and clear. 

I have two minor comments:

(1) In the introduction, the authors mention that the COTiD intervention has been adapted to make it more suitable to the UK. There however is no mention of what these adaptations are. This has probably been described in the protocol paper the authors refer to, but in view of trying to understand why the intervention does not have the intended effect, I think there are good reasons for (re)listing these adaptations here, and to assess whether these adaptations may have had an impact on the results and be part of the explanation of why the results in the Netherlands were different.

(2) At the end of the paper, when comparing the study to similar studies in other countries, the authors state that usual care in the UK often encompasses occupational therapy. Have the participants been asked if and how much OT they received before the intervention? If so, should the number of hours receiving occupational therapy not be mentioned in table 1?

[LINK]

---

## [Decision Letter · Decision Letter 2]

2 Oct 2020

Dear Dr. Wenborn,

Thank you very much for re-submitting your manuscript "Community Occupational Therapy for people with dementia and family carers (COTiD-UK) versus treatment as usual (Valuing Active Life in Dementia [VALID]) study: a single-blind, randomised controlled trial" (PMEDICINE-D-20-00486R2) for review by PLOS Medicine.

I have discussed the paper with my colleagues and the academic editor and it was also seen again by one of the original reviewers. I am pleased to say that provided the remaining editorial and production issues are dealt with we are planning to accept the paper for publication in the journal.

[LINK]

We look forward to receiving the revised manuscript by Oct 09 2020 11:59PM. 

Sincerely,

Thomas McBride, PhD

Senior Editor 

PLOS Medicine

plosmedicine.org

Requests from Editors:

1- Regarding your trial protocol, the reference is sufficient, no need to add a link in the main text.

2- Thank you for agreeing to make your data available. At this time, please provide the link to the data repository and accession numbers required for access. Please provide the data statement in the appropriate metadata section, and remove the statement from the main text file.

3- Thank you for providing your CONSORT checklist. Please replace the page numbers with paragraph numbers per section (e.g. "Methods, paragraph 1"), since the page numbers of the final published paper may be different from the page numbers in the current manuscript.

4- Thank you for editing the Abstract Sections. Please reformat the text so that each section is one paragraph.

5- You can remove the two sentences on lines 92-94 of the Abstract “All participants had to be able to communicate in English...”

6- Please include the dates of the study and some demographic information on the study participants (age, sex) in the Abstract Methods and Findings.

7- The statement on line 107 (“91% of the goals set…”) could use a bit more context. Were the goals for performing certain tasks, and what were the timeframes? Alternatively, this could be left out of the Abstract as it does not seem to be one of the registered outcomes.

8- The last sentence of the Abstract Conclusions is a bit confusing, why are the study outcomes (ADL, cognition, QOL, etc) not meaningful? The phrase rely less on proxy data" could be removed or rephrased to be more specific. This also applies to the last point of the Author Summary.

9- Thank you for adding an Author Summary. The first section should include a point that lays out the knowledge gap (e.g., developing effective strategies for delivering OT to people with dementia). The first two points of the second section could make it clearer that the pairs were randomized to either usual treatment of the COTiD-UK program.

10- Please rewrite the final section of the Author summary to more clearly explain the new knowledge generated by the research and specific implications for practice, research, policy, or public health.

11- Please move the sections mentioning the CONSORT statement and checklist into the Methods section.

12- Please describe Fig 1 as “Participant flowchart” and reformat to remove the black background.

13- Please reorganize the Discussion as follows: a short, clear summary of what was done and the article's findings; what the study adds to existing research and where and why the results may differ from previous research; strengths and limitations of the study; implications and next steps for research, clinical practice, and/or public policy; one-paragraph conclusion.

14- The current concluding paragraph “There is a need for innovation and improvement in dementia care…” is better suited to the penultimate section “ implications and next steps for research” of the Discussion.

15- Please remove the Funding statement and Competing interests statement from the main text into the corresponding metadata sections.

16- Please move the ethical approval section from the end of the manuscript to the Methods section.

Comments from Reviewers:

Reviewer #2: Thanks for the revised manuscript and the inclusion of the SAP. All of my original queries have been resolved thoroughly (and I agree with your point about marginal estimates in the graph), and I only have a few minor queries. 

The SAP was very helpful - particularly the details around the main analysis and the CACE. Seeing the model as an equation was much easier for me than parsing the text. Could the details about the main analysis and the CACE be included in the supplementary appendixes (or maybe include a link to the SAP in a repository)? I think that you have made a great approach to this type of data and this could be a helpful reference for anyone faced with similar data in future. Writing a SAP for a complex multi-level model isn't an easy undertaking and I think this a good example for anyone trying to do the same.

 What was used to test the difference between homogenous vs. heterogeneous patient-level models, and which was selected in the main paper?

One last query from the SAP - does the 75% and 25% of the sample at 52 and 78 weeks refer to a sampling fraction from the original survey, or a quota sample?

[LINK]

---

## [Editor Report · Decision Letter 3]

24 Nov 2020

Dear Dr. Wenborn, 

On behalf of my colleagues and the academic editor, Dr. Carol Brayne, I am delighted to inform you that your manuscript entitled "Community Occupational Therapy for people with dementia and family carers (COTiD-UK) versus treatment as usual (Valuing Active Life in Dementia [VALID]) study: a single-blind, randomised controlled trial" (PMEDICINE-D-20-00486R3) has been accepted for publication in PLOS Medicine. 

PRODUCTION PROCESS

Before publication you will see the copyedited word document (within 5 business days) and a PDF proof shortly after that. The copyeditor will be in touch shortly before sending you the copyedited Word document. We will make some revisions at copyediting stage to conform to our general style, and for clarification. When you receive this version you should check and revise it very carefully, including figures, tables, references, and supporting information, because corrections at the next stage (proofs) will be strictly limited to (1) errors in author names or affiliations, (2) errors of scientific fact that would cause misunderstandings to readers, and (3) printer's (introduced) errors. Please return the copyedited file within 2 business days in order to ensure timely delivery of the PDF proof. 

If you are likely to be away when either this document or the proof is sent, please ensure we have contact information of a second person, as we will need you to respond quickly at each point. Given the disruptions resulting from the ongoing COVID-19 pandemic, there may be delays in the production process. We apologise in advance for any inconvenience caused and will do our best to minimize impact as far as possible.

EARLY VERSION

PRESS

PROFILE INFORMATION

Thank you again for submitting the manuscript to PLOS Medicine. We look forward to publishing it. 

Best wishes, 

Thomas McBride, PhD

Senior Editor 

PLOS Medicine

plosmedicine.org